# Red meat consumption and its association with hypertension and hyperlipidaemia among adult Maasai pastoralists of Ngorongoro Conservation Area, Tanzania

Ester J. Diarz[1,2]*, Beatrice J. Leyaro[1], Sokoine L. Kivuyo[2], Bernard J. Ngowi[2,3], Sia E. Msuya[1,4,5], Sayoki G. Mfinanga[2,6,7], Bassirou Bonfoh[8], Michael J. Mahande[1]

1 Department of Epidemiology and Applied Biostatistics, Institute of Public Health, Kilimanjaro Christian Medical University College, Kilimanjaro, Tanzania, 2 Muhimbili Medical Research Centre, National Institute for Medical Research, Dar es salaam, Tanzania, 3 Mbeya College of Health and Allied Sciences, University of Dar es salaam, Mbeya, Tanzania, 4 Department of Community Health, Institute of Public Health, Kilimanjaro Christian Medical University College, Kilimanjaro, Tanzania, 5 Department of Community Medicine, Kilimanjaro Christian Medical Centre, Kilimanjaro, Tanzania, 6 The Muhimbili University of Health and Allied Sciences, Dar es salaam, Tanzania, 7 Liverpool School of Tropical Medicine, Liverpool, United Kingdom, 8 Centre Suisse de Recherches Scientifiques en Côte d'Ivoire, Abidjan, Côte d'Ivoire

* diarzester@gmail.com

## Abstract

### Background

Red meat is an important dietary source of protein and other essential nutrients. Its high intake has been associated with an increased risk of cardiovascular morbidity and mortality, including hypertension (HTN) and hyperlipidaemia (HLP). Despite being physically active, the Maasai at Ngorongoro Conservation Area (NCA) depend heavily on animals' products as their staple food with fewer intakes of vegetables or fruits due to restriction from carrying out agricultural activities within the NCA. This study aimed at determining the prevalence of HTN and HLP and their association with red meat consumption among adult Maasai of NCA.

### Methods

A community-based cross-sectional study was conducted in October 2018 using multistage sampling technique. Eight hundred and ninety-four (894) participants enrolled from seven villages in three wards within NCA Data were collected using a modified WHO NCDs-STEPS tool. Anthropometric measurements, blood pressure (BP) measurements, and blood samples for glucose and cholesterol tests were obtained from the study participants. Crude and adjusted prevalence ratio (PR) for factors associated with HTN and HLP were estimated using Ordinal and Bayesian logistic regression models, respectively.

### Results

The prevalence of HLP was 23.7 percent. The levels were higher among males than were among the females (29.0% vs. 20.1%, p = 0.002). The prevalence of HTN and pre-HTN (elevated BP) were 9.8 and 37.0 percent, respectively. Both HTN and elevated BP were higher

**Data Availability Statement:** Dataset is publicly available and uploaded to the Figshare repository (DOI:10.6084/M9.FIGSHARE.12345986).

**Funding:** EJD sponsored by AfriqueOne ASPIRE and the study was conducted within the framework of the DELTAS Africa Initiative [Afrique One-ASPIRE /DEL-15-008]. Afrique One-ASPIRE is funded by a consortium of donor including the African Academy of Sciences (AAS) Alliance for Accelerating Excellence in Science in Africa (AESA), the New Partnership for Africa's Development Planning and Coordinating (NEPAD) Agency, the Welcome Trust [107753/A/15/Z] and the UK government. http://afriqueoneaspire.org The funder had no role in study design, data collection and analysis, decision to publish, or preparation of the manuscript.

**Competing interests:** The authors have declared that no competing interests exist.

among males than were among females (hypertensive [10.9% vs. 9.0%]; elevated BP [44.0% vs. 32.1%], p<0.001). The prevalence of HLP was significantly associated with level **II** (PR = 1.56, 95%CrI: 1.10–2.09) and level **III** (PR = 1.64, 95%CrI: 1.08–2.41) of red meat consumption as opposed to level **I**.

## Conclusion

The prevalence of hyperlipidaemia and elevated BP were high among NCA Maasai. We found a significant association between red meat consumption and hyperlipidaemia. Further follow-up studies are warranted to establish a temporal relationship between red meat consumption and both conditions.

## Introduction

Cardiovascular diseases (CVDs) are the number one -cause of mortality globally. The CVDs are increasingly becoming prevalent in low and middle-income countries (LMICs) due to an increase of risk factors such as high blood pressure and overweight/obesity [1]. Both hypertension (HTN) and hyperlipidaemia (HLP) increase the risk of CVDs, particularly heart diseases and stroke [1]. HTN is a leading risk factor for CVDs, while HLP is among the most important modifiable risk factors for myocardial infarction and ischemic stroke [2,3]. HLP is reported to be killing approximately 9.4 million people annually worldwide and accounts for 57 million disability-adjusted life years (DALYs) (1,4). Overall, HLP is estimated to be causing 2.6 million deaths and 29.7 million DALYS [4]. In a meta-analysis, which included 242 studies, conducted in middle-income countries (MICs) projected that more than 75 percent of the hypertensive populations would be living in LMICs by 2025 [5]. In Tanzania, CVDs accounts for 13percent among the 409,000 total deaths caused by NCDs annually [6]. A study which was carried out among the agro-pastoral community in Tanzania (Simanjiro-Arusha) reported the overall prevalence of HTN of 21.4 percent [7]. However, little has been documented on HLP among the Maasai pastoralists in Tanzania.

If HTN and HLP remain uncontrolled, they may cause several disorders among the people such as Ischemic stroke, myocardial infarction, cardiac failure, dementia, renal failure and blindness [1]. Both HTN and HLP are likely to increase the burden of diseases and management costs at both national and family levels due to complications associated with the presence of these conditions [8,9]. However, the control of these conditions is linked to a change in lifestyle and dietary habit such as the control of red meat consumption.

Red meat has long been established as an important dietary source of protein and essential nutrients including iron, zinc, and vitamin B12 [9]. High intake of red meat has also been associated with a higher risks of cardiovascular morbidity and mortality, such as HTN and HLP [10]. Previous studies have revealed that the relationship between meat consumption and development of these conditions may vary depending on the extent of meat processing (i.e., unprocessed/fresh meat or processing meat by adding high amounts of salt, frying or drying) [11,12]. Because of the linkage between red meat consumption and increased risk of CVDs, the Department of Health in the United Kingdom (UK) issued new guidelines in 2011 on eating and processing red meat. The guidelines require that for adults who eat more than 90 g of red and processed meat per day should reduce their intake to an average of 70 g/day (cooked weight). The current average consumption of red and processed meat is 86 g/day among men and 56 g/day among women [13].

Traditionally, Tanzanian Maasai used to roam with their cattle throughout in northern Tanzania due to their nomadic lifestyle to search for grazing resources such as pastures and water [14,15]. Ngorongoro Conservation Area (NCA) is a unique protected area which integrates the conservation of soils, vegetation, wildlife, and watersheds in ecosystem that involve sustainability of pastoralism and the tourist industry [16]. The NCA has sustained its natural ecosystem as a result many Maasai communities have moved to this area to exploit the natural resources in practising their traditional herding systems [17]. However, there is a regulation, which was instituted by the NCA Authority restricting residents (the Maasai) against carrying out agricultural activities within the NCA. The regulation was instituted to protect the existing human-animal ecosystem and the flourishing tourism industry [16]. To enforce this regulation, the Government has been distributing maize to the Maasai community in the NCA, to enable them have a balanced diet. However, it is reported that the government efforts have had little impact because the Maasai communities cannot afford to buy maize even at the subsidized price [18].

Despite being physically active [19], the NCA Maasai are reported to have a different pattern of NCDs risk. Unavailability of land for grazing and their nomadic lifestyle influence their dietary lifestyle into dependence almost entirely on animal-based products with an inadequate intake of vegetables and fruits or roots tubers something that limits food diversity. In addition, the Maasai residing in NCA have inadequate access to health facilities. These are normally not well equipped to manage HTN and HLP. Furthermore, their knowledge on a healthy diet and NCDs risk in general, is limited. These multiple factors put NCA Maasai at higher risk of NCDs, including HTN and HLP. However, information on the burden of HTN and HLP and its association with red meat consumption among the Maasai at NCA is scant.

This study aimed to determine the prevalence of HTN and HLP and their association with red meat consumption among adult Maasai communities in NCA, in Tanzania. This information is essential to health practitioners, policymakers, and program planners in developing focused policies that would work best in the pastoral community such as the NCA in controlling and preventing both conditions.

## Material and methods

### Study design and site

This study was a community-based cross-sectional and was conducted in October 2018 within the Ngorongoro Conservation Area (NCA)-Arusha region, northern Tanzania. NCA is a protected area and a World Heritage site, which was established in 1959 and designated as Game Reserves and National Parks. It had a population of approximately 50,000 people in 2012 [15], covering approximately 8,292 square kilometres. NCA is located 180 km in the Crater Highlands west of Arusha [20]. The NCA authority allows human habitation (the majority of who are Maasai pastoralists) in the protected area, despite that livelihood activities are restricted in the NCA, including cultivation and livestock grazing except watering of cattle. As a result of the population increase, in 2009, the government of Tanzania announced the imminent eviction of some 30,000 local Maasai, reduction of the NCA authority staff and facilities from within the NCA to a location outside the NCA and the immediate ban of all forms of cultivation including home gardens [15,20]. This restriction prevents the emergence of adverse effects on wildlife population. the main economic activity of NCA residents is pastoralism which is currently on a decline but complimented by tourism activities (cultural boma) [21]. Most of the health facilities within NCA are dispensaries with one hospital owned by Catholic Archdiocese of Arusha in collaboration with the Government of Tanzania under the Ngorongoro District Council. The hospital provides healthcare to both outpatient and inpatients with an

average of 60 outpatients per day, the hospital administers an average of 18 outreach clinics a month. The clinks are mainly for reproductive and child health clinic. It is equipped with x-ray machine, ultrasound machine and laboratory equipment [22].

## Study population, sample, size, and sampling procedures

The study enrolled participants from the Maasai tribe (ethnic) with at least 18 years of age who were able to respond to study questions (either verbally or through a translator), who provided voluntary informed consent and residing in the study area (within the selected villages) in the NCA. Pregnant mothers were excluded in the study. A minimum estimated sample size was 960 participants based on the prevalence of hypertension of 21.3 percent recorded in a previous study conducted in 2015, in Simanjiro district, Arusha region, northern, Tanzania [7]. A standard normal value (1.96) at 95 percent confidence interval, 3 percent margin of error (e), and 25 percent probability of non-responses was used. The formula for estimating a single population proportion with specified absolute precision was used [23], and the study sample was drawn using a multistage sampling technique. The NCA was purposely selected because of its uniqueness of allowing human habitation but restricting cultivation. Three wards were randomly selected among six wards of NCA. Seven villages namely, Enduleni, Ndian, and Nasporiong from Endulen ward; Essere and Laitole villages from Alaitole ward; and Misigiyo and Kaitakiteng' villages from Misigiyo ward were randomly selected. A loudspeaker was used to around all the selected villages to make announcement and extend invitation to all villagers to participate in the recruitment. The verbal announcement was made between 7–8 pm before the interview date and was repeated for two consecutive days. Participants were asked to gather either at village office or school depending on the comfortability of the area on a specified day. The participants who showed up and agreed to participate in the study were screened for eligibility and enrolled. The same procedures were repeated in each village visited. For the case of Moran Maasai (These are Maasai boys of between 18 to 25 years who underwent circumcision ceremony performed without anaesthetic), who were mostly not at home during the afternoon because of their cultural responsibility of herding cattle, were visited at night from 7 pm for interviews.

Data collection tool was adopted from WHO Non-Communicable Diseases Stepwise approach to surveillance (STEPS) https://www.who.int/ncds/surveillance/steps/en/ This was then modified with additional questions to account for cultural diet of Maasai in Tanzania (attached in supporting information in both Swahili and English language). The questionnaire included; social demographic information (age gender, income etc), data on life style (physical activity, tobacco use, alcohol consumption, dietary intake) and relevant family history information was collected from each participant through self-reporting. The quantity of red meat consumed, anthropometric, and BP were measured. Samples of blood glucose and the total cholesterol levels were collected and tested. Training of the data collection activity was conducted for five days. Pre-test of the tools and procedures were done in 4th day of training to the community around Kilimanjaro Christian Medical University College area. After the pre-test, feedback was shared, challenges observed from the pre-test were explained and finalization of the tools were done. The data collection team included trained senior research scientists, year-three laboratory students, and year-five medical students.

## Red meat consumption and parameters measurements

**Quantity of red meat consumed.**   Red meat consumptions were measured through self-reporting by the participant through showing participants the pictures of utensils to measure the amounts. The utensils were used to determine the amounts of red meat consumed in a

typical meal. Each utensil has an average size in a millilitre of water, which is equivalent to the gram in adult meat portion measurement charts converter. Hence, the amount of red meat consumed was estimated using adult meat portion measurement charts after self-reporting. Both the pictures of the utensil and adult meat portion measurement charts were adopted from Tanzania food composition table [24]

**Anthropometric measurements.** Two trained medical students (years 5) measured the participants' anthropometric (height, weight, waist, and hip circumference). Height was measured to the nearest 0.1 centimetres, and weight to the nearest 0.1 kilograms; these measurements were taken while participants were wearing light clothes and without shoes and using a locally available Stadiometer and weighing scale (Seca, CA, USA) respectively. Waist and hip circumferences were measured (in centimetres) around the widest portion of the buttocks with the tape parallel to the floor. Body Mass Index (BMI) was calculated as weight (kilograms) divided by height (meters) squared ($kg/m^2$) and categorized as follows underweight ($<18.5kg/m^2$), normal ($18.5–24.9kg/m^2$), overweight ($25–29.9kg/m^2$), and obese ($\geq30kg/m^2$). The waist to hip ratio (WHR) was calculated as waist (centimetres) divided by hip (centimetres) and categorized as follows normal [$\leq0.90$ (males) and $\leq0.85$ (females)] and abnormal [$>0.90$(males) and $>0.85$(females)].

**Blood pressure estimate.** BP was measured (in mmHg) to every participant's right arm on a seating position using an automatic blood pressure cuff (Riester E-Mega, Germany). The measurements were taken by a trained study doctor/year-five medical student. Before taking the BP measurements, the participants were allowed to rest for 5 minutes. The measurements were recorded three times of at least 10 minutes time interval between BP readings. The mean of the three readings was calculated and blood pressure was classified as follows normal ($<120/80mmHg$), elevated BP ($120/80mmHg-139/89mmHg$), and hypertensive ($\geq140/90mmHg$ or self-reported use of anti-hypertensive medications).

**Blood glucose level test.** Most of the participants were able to test for random blood glucose (RBG). The fasting blood glucose (FBG) test was difficult to measure because most of the participants had already eaten something before testing blood glucose. This is because the sample for the FBG is measured after at least 8 hours of overnight fasting. However, those with RBG of 11.1mmol or higher were requested to come for FBG the following day. The level of blood glucose was determined using Glucometer (Gluco Plus $^{TM}$, Canada) by a trained year-three laboratory science student at the site. A small drop of the blood sample was obtained from a participant's finger by pricking the finger with a lancet, placed on a disposable test strip where the meter reads and uses to calculate the blood glucose level. The results were ready and displayed in mmol/L after 5 to 40 second. A senior laboratory technician verified the results. The readings for FBG of >7mmol/L and RBG of >11mmol/L were considered as raised blood glucose.

**Total cholesterol level test.** The level of serum total cholesterol was determined by a rapid test at the site using cholesterol machine (Accutrend $^{TM}$ plus). The devices were validated and calibrated by a trained Senior Laboratory Technician. A small drop of the blood sample was obtained from a participant's finger by pricking the finger with a lancet, placed on a disposable test strip where the device reads and calculates the total cholesterol in the blood sample. After 180 seconds wait (or after a beeper was enabled) the results were automatically displayed in mg/dL. A Senior Laboratory Technician verified the results. The readings for the total cholesterol $\geq200mg/dL$ were considered as HLP.

## Data management and statistical analysis

Data were collected using Open Data Kit (ODK collect version 1.22.1). the collected data were then transferred to STATA version 15.1 statistical software (StatCorp, College Station, TX) to enhance data cleaning and analysis. Descriptive statistics were summarized using frequencies

and proportions for categorical data while the measure of central tendency with their respective statistical measure of dispersion was used to summarize continuous data. Chi-square ($\chi^2$) statistic was used to determine the difference between the sets of covariates and outcomes of interest (HLP and HTN). Ordinal logistic regression was used to determine the association between HTN and red meat consumption. The prevalence of HLP was more than 10percent; hence, a classical logistic regression was thought to be an inappropriate approach to model the outcome. The Bayesian logistic regression was considered as the best alternative. This approach provided a flexible framework for modelling common outcomes. This model includes a prior distribution component of the covariates and produces 95percent Credible Intervals (CrI) for the estimates and gives a better inference [25,26]. Bivariate Ordinal and Bayesian logistic regression analyses were used in adjusting for the potential confounders of the association between HTN and HLP and red meat consumption. A variable was considered a potential confounder if it leads to a change of the model in PR of the primary exposure by 10percent. During crude analyses, variables that were significantly associated with either outcome, but were not confounders, were noted as independent predictors of that outcome. Both confounders and independent predictors were included in the multivariable models to adjust for any possible confounding and residual confounding. The selection of the model with better goodness of fit was made using Akaike's Information Criterion and Deviance Information Criterion for both HTN and HLP, respectively. The PRs with 95percent CI and 95percent CrI that does not include null for HTN and HLP were considered as statistically significant.

## Ethical consideration

Ethical approval was obtained from the Research Ethical Committee of Kilimanjaro Christian Medical University College (CREC no. 2387). The goals and benefits of the study were explained to the participants. The written consents or thumbprints (for the participants who were unable to write) were obtained from the participants as appropriate for the cultural and literacy considerations. The consents were sought and structured questions were administered in the local Kiswahili language. For the participants who spoke the Maasai language only, native Maasai speaker who was a member of the research support team and a trained medical officer helped to translate the scripts for them to comprehend information and participate in the study. Information on the physical and biochemical examinations was kept confidential and was shared with the client and clinical personnel only. All the participants received individual feedback on the results of their examination and, where necessary, they were referred to Endulen Hospital for appropriate follow-up. All information obtained from the participants was kept and analysed anonymously.

## Results

### Socio-economic characteristics of the study participants

Eight hundred and ninety-four (894) participants were enrolled and analysed in this study. The median age (IQR) was 35 (26–50) years. About 30percent of the participants were aged between 25–34 years. More than half (59.1%) were females, and the majority (64.2%) had no formal education. The majority (80.6%) of the participants were married and 77.0 percent were engaged in pastoral activities (**Table 1**).

### Behavioural, dietary, and biological characteristics of the participants

The majority (99%) of the study participants were consumers of red meat. Their mean (SD) body mass index was 21.9 (3.9) kg/m$^2$, and the ratio of the waist to hip was 0.9 (0.1). The

**Table 1. Socio-economic characteristics of the study participants (N = 894).**

| Characteristics | N | (%) |
|---|---|---|
| **Age (years)** | | |
| [Median (IQR)] | 35(26–50) | |
| 24 or less | 176 | (19.7) |
| 25 to 34 | 263 | (29.4) |
| 35 to 44 | 152 | (17.0) |
| 45 to 54 | 129 | (14.4) |
| ≥55 | 174 | (19.5) |
| **Sex** | | |
| Female | 528 | (59.1) |
| Male | 366 | (40.9) |
| **Marital Status** | | |
| Never married/Separated | 173 | (19.4) |
| Currently married/Cohabiting | 721 | (80.6) |
| **Education level** | | |
| No formal school | 574 | (64.2) |
| Primary school | 237 | (26.5) |
| Secondary school or above | 83 | (9.3) |
| **Villages** | | |
| Endulen | 175 | (19.6) |
| Essere | 49 | (5.5)) |
| Laitole | 129 | (12.2) |
| Misigiyo | 132 | (14.4) |
| Kaitakiteng' | 109 | (14.8) |
| Nasporiong' | 179 | (20.0) |
| Ndian | 121 | (13.5) |
| **Occupation status past 12 months** | | |
| No job/student | 113 | (12.6) |
| Pastoralist | 688 | (77.0) |
| Business/Professional Employed | 93 | (10.4) |
| **Income per month (TZS)[#]** | | |
| [Median (IQR)] | 80,000(40,000–200,000) | |
| No Income | 51 | (5.7) |
| ≤100,000 | 156 | (17.5) |
| > 100,000 | 170 | (19.0) |
| Did not disclose | 517 | (57.8) |
| **Number of people in household ≥18years** | | |
| [Median (IQR)] | 4(2–6) | |
| ≤5 people | 653 | (73.0) |
| >5 people | 241 | (27.00) |

[#]1USD = 2,270TZS

average mean (SD) systolic and diastolic BP were 117.6 (15.2) mmHg and 75.4 (10.2) mmHg, respectively, while, the mean (SD) total cholesterol was 177.8 (38.9) mg/dL (Table 2).

## Level of meat consumption

Most (99.4%) of the participants reported being consumers of red meat. Half (50.8%) of these reported to have been consuming red meat within the recommended threshold (500 grams/

**Table 2. Behavioural, dietary, and biological characteristics of study participants (N = 894).**

| Characteristics | N | % |
|---|---|---|
| **Currently smoke/sniff/chew any tobacco products** | | |
| No | 681 | (76.2) |
| Yes | 213 | (23.8) |
| **Currently, in the past 12 months consume any alcohol** | | |
| No | 720 | (80.5) |
| Yes | 174 | (19.5) |
| **Red meat consumption** | | |
| No | 5 | (0.6) |
| Yes | **889** | **(99.4)** |
| **Ingest cow's blood** | | |
| No | 644 | (72.0) |
| Yes | 250 | (28.0) |
| **Form of blood ingested (N = 250)** | | |
| Fresh | 181 | (72.4) |
| Processed | 29 | (11.6) |
| Both | 40 | (16.0) |
| **Frequency of ingesting animal (cow) blood (N = 250)** | | |
| Occasionally | 151 | (60.4) |
| Often | 90 | (36.0) |
| Always | 9 | (3.6) |
| **Frequency of adding animal fats to the food right before eating** | | |
| Never | 273 | (30.5) |
| Rarely | 272 | (30.4) |
| Sometimes | 349 | (39.0) |
| **Weekly physical activities (work/walk/bicycle/sport)** | | |
| No | 23 | (2.6) |
| Yes | 871 | (97.4) |
| **Family history of any CVDs from 1st degree relative** | | |
| No | 853 | (95.4) |
| Yes | 41 | (4.6) |
| **Body Mass Index (kg/m$^2$)** | | |
| [Mean (SD)] | 21.9 (3.9) | |
| Normal (18.5–24.9) | 619 | (69.2) |
| Underweight ($\leq$18.4) | 134 | (15.0) |
| Overweight (25–29.9) | 108 | (12.1) |
| Obese ($\geq$30) | 33 | (3.7) |
| **Waist to Hip Ratio (abnormal: >0.90(males) & >0.85(females)** | | |
| [Mean (SD)] | 0.9(0.1) | |
| Normal | 390 | (43.6) |
| Abnormal | 504 | (56.4) |
| **Blood glucose level in mmol/L (RBG>11 or FBG>7 = Raised)** | | |
| [Mean (SD)] | [5.2(1.3)]¥[5.3(2.7)]* | |
| Normal | 882 | (98.7) |
| Raised blood glucose | 12 | (1.3) |
| **Total cholesterol in mg/dl** | | |
| [Mean (SD)] | 177.7(38.9) | |
| Normal | 682 | (76.3) |

(*Continued*)

**Table 2.** (Continued)

| Characteristics | N | % |
|---|---|---|
| Raised TC | 212 | (23.7) |
| **Average SBP in mmHg** [Mean (SD)] | 117.6 (15.2) | |
| **Average DBP in mmHg** [Mean (SD)] | 75.4 (10.2) | |

¥ Random blood glucose [RBG] (total number = 853)

*Fasting blood glucose [FBG] (total number = 41)

SBP = Systolic Blood pressure; DBP = Diastolic Blood pressure.

week). Majority (61%) of females reported to have been consuming red meat within the recommended threshold as opposed to their male counterparts (37%); p<0.001(**Fig 1**).

## Prevalence of HTN and HLP by levels of red meat consumption

The overall prevalence of elevated BP was high in all three levels of red meat consumption followed by the prevalence of HLP and HTN (**Fig 2**). The prevalence of elevated BP increased from level **I** to level **II** of red meat consumption [≤500 g/week (35.0%) to 501–1000 g/week (40.6%) respectively] then it declined in level **III** (>1000 g/week) to 37.8percent. In contrast, the prevalence of HTN decreased with an increase of the consumption of red meat [≤500 g/week (10.4%); 501–1000 g/week (9.6%), and >1000 g/week (8.7%)]. However, this difference was not statistically significant (p = 0.669). On the other hand, the prevalence of HLP slightly increased with an increase in the levels of red meat consumption [≤500 g/week (21.4%); 501–1000 g/week (26.1%) and >1000 g/week (26.5%); p = 0.235].

## Prevalence of HLP and HTN by socio-economic characteristics

Overall, there was 23.7 percent prevalence of HLP. This prevalence was significantly higher among males than was among females [29.0% vs. 20.8%, respectively, p = 0.003]. There was a variation in the prevalence of HLP across age groups as follows ≤24 years (14.0%25–34 years

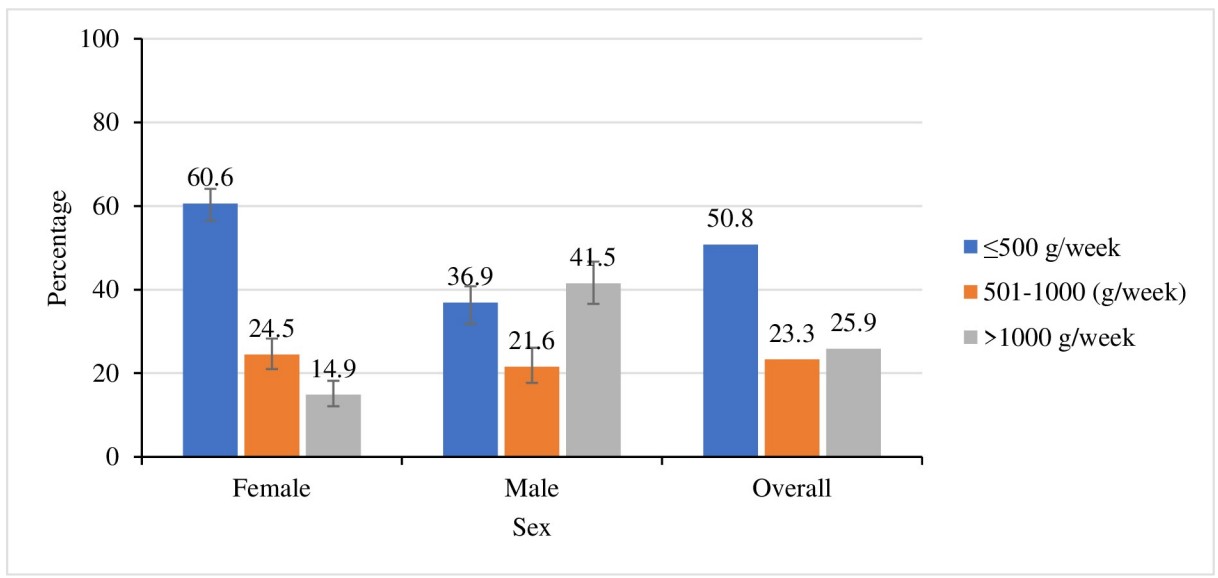

**Fig 1. Percentage of the level of red meat consumption by sex.** The Observed difference between sexes was statistically significant at p<0.001.

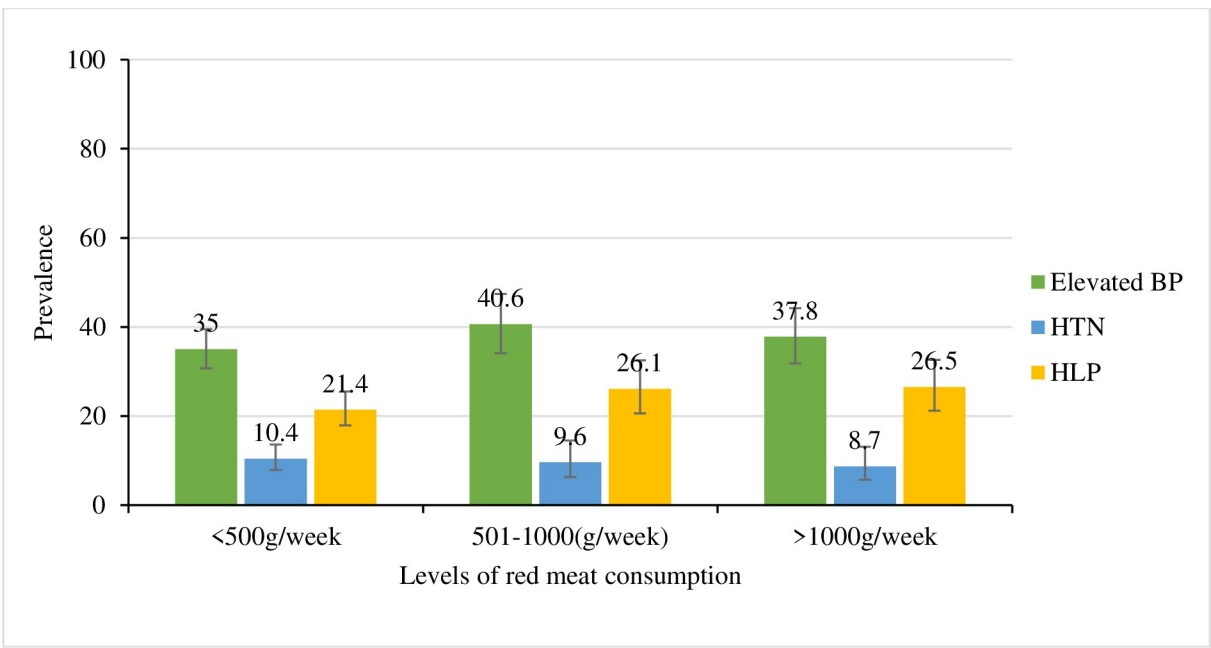

**Fig 2. Prevalence of HLP and HTN by levels of red meat consumption.** The observed difference was not statistically significant.

(24%); 35–44 years (25%); 45-54years (24%); and ≥55 years (32%). The variation in the prevalence of HLP across age groups was statistically significant at p = 0.002 (Table 3).

On the other hand, the overall prevalence of HTN was 9.8percent while that of the elevated BP was 37.0percent. Both hypertensive and elevated BP were significantly higher among males than was among females [(10.9% and 44.0%) vs. (9.0% and 32.1%), respectively, p<0.001]. The prevalence of HTN was significantly different across age groups, with higher prevalence (16.8%) among older individuals (≥55 years) than was the case with young counterparts (4.6%) (p = 0.001) (Table 3).

Participants earning a monthly income of ≥100,000TZS had higher prevalence of HLP (29.9%; p = 0.003) {unlike the case with HTN (36.5%; p = 0.468)} than is the case with their counterparts who earned less in a month. The higher prevalence of HLP (>20%) was observed among participants residing in Endulen, Misigiyo and Kaitakiteng' as opposed to other villages. This difference was not statistically significant for HTN. Similarly, there was no statistically significant difference in the prevalence of both HLP and HTN across levels of education, types of occupation, sizes of households, and marital status (Table 3).

## Prevalence of HTN and HLP by behavioural characteristics

Participants who reported to have been using any tobacco products had a significantly high prevalence of HTN as opposed to those who reported not using tobacco (15.5% vs. 8.0%; p = 0.001). There was no difference in the prevalence of HLP between tobacco users and non-users (24% vs. 23.5%; p = 0.884). Participants who reported to have had consumed alcohol in the previous 12 months had a significantly higher prevalence of both HTN and HLP as compared to their counterparts who had consumed alcohol [(20.1% vs. 7.3%; p<0.001) and (34.5% vs. 21.3%; p<0.001), respectively]. Participants who reported to have been engaged in performed any physical activity (vigorous, intensity, sport or walking) had lower prevalence of HLP as compared to their counterparts who had not been engaged in any physical activities (23.4% vs. 40.9%; p = 0.078). However, this was not the case for HTN (Table 4).

**Table 3. Prevalence of HTN and HLP by socio-economic characteristics (N = 889).**

| Characteristics | Total | Blood Pressure | | p-value for trend | HLP$^{\phi}$ | $\chi^2$ p-value |
|---|---|---|---|---|---|---|
| | | Elevated-BP | HTN* | | | |
| | | n = 329 (37.0%) | n = 87(9.8%) | | n = 212 (23.7%) | |
| **Sex** | | | | | | |
| Female | 523 | 168(32.1) | 47(9.0) | <0.001 | 106(20.3) | 0.003 |
| Male | 366 | 161(44.0) | 40(10.9) | | 106(29.0) | |
| **Age (years)** | | | | | | |
| 18 to 24 | 173 | 63(36.4) | 8(4.6) | 0.001 | 24(13.9) | 0.002 |
| 25 to 34 | 263 | 99(37.6) | 17(6.5) | | 63(24.0) | |
| 35 to 44 | 152 | 51(33.6) | 16(10.5) | | 38(25.0) | |
| 45 to 54 | 128 | 43(33.6) | 17(13.3) | | 31(24.2) | |
| ≥55 | 173 | 73(42.2) | 29(16.8) | | 56(32.4) | |
| **Marital Status** | | | | | | |
| Never married/separated | 172 | 66(38.4) | 20(11.6) | 0.533 | 35(20.3) | 0.231 |
| Married/Cohabiting | 717 | 263(36.7) | 67(9.3) | | 177(24.7) | |
| **Villages** | | | | | | |
| Endulen | 175 | 66(37.7) | 17(9.7) | 0.769 | 52(29.7) | <0.001 |
| Essere | 48 | 18(37.5) | 7(14.6) | | 8(16.7) | |
| Laitole | 129 | 45(41.3) | 12(11.0) | | 15(11.6) | |
| Misigiyo | 132 | 42(32.6) | 9(7.0) | | 49(37.1) | |
| Kaitakiteng' | 109 | 43(32.6) | 15(11.4) | | 31(28.4) | |
| Nasporiong' | 176 | 69(39.2) | 18(10.2) | | 35(19.9) | |
| Ndian | 120 | 46(38.3) | 9(7.5) | | 22(18.3) | |
| **Current education level** | | | | | | |
| No formal school | 569 | 211(37.1) | 56(9.8) | 0.837 | 124(21.8) | 0.101 |
| Primary school | 237 | 83(35.0) | 23(9.7) | | 62(26.2) | |
| ≥Secondary school | 83 | 35(42.2) | 8(9.6) | | 26(31.3) | |
| **Occupation status past 12 months** | | | | | | |
| No job/student | 112 | 31(27.7) | 9(8.0) | 0.149 | 21(18.8) | 0.287 |
| Pastoralist | 684 | 265(38.7) | 68(9.9) | | 165(24.1) | |
| Business/Employed | 93 | 33(35.5) | 10(10.8) | | 26(28.0) | |
| **The income per month (TZS)** | | | | | | |
| No Income | 51 | 20(39.2) | 3(5.9) | 0.468 | 7(13.7) | 0.003 |
| ≤100,000 | 154 | 49(31.8) | 15(9.7) | | 23(14.9) | |
| > 100,000 | 167 | 68(40.7) | 12(7.2) | | 50(29.9) | |
| Did not disclose | 517 | 192(37.1) | 57(11.0) | | 132(25.5) | |
| **Number of people in HH ≥18years*** | | | | | | |
| ≤5 people | 648 | 235(36.3) | 61(9.4) | 0.534 | 160(24.7) | 0.333 |
| < 5 people | 241 | 94(39.0) | 26(10.8) | | 52(21.6) | |

*<120/80mmHg = Normal BP, 120/80-139.9/89.9mmHg = Elevated BP, ≥140/90mmHg = HTN

$\phi$<200mg/dL = Normal TC levels, ≥200mg/dL = HLP

## Prevalence of HTN and HLP by biological characteristics

Participants with raised cholesterol levels had a significantly higher prevalence of HTN as compared to those who had normal cholesterol levels (17.5% vs. 7.4%; p<0.001) (**Not shown**). Participants who reported to have had a family history of any CVDs from their first degree relative had higher prevalence of both HTN and HLP as compared to their counterparts who did

**Table 4. Prevalence of HTN and HLP by participants' behavioural and biological characteristics (N = 889).**

| Characteristics | Total | Blood Pressure | | p-value for trend | HLP$^\phi$ | $\chi^2$ p-value |
|---|---|---|---|---|---|---|
| | | Elevated-BP | HTN* | | | |
| | | n = 329 (37.0%) | n = 87 (9.8%) | | n = 212 (23.7%) | |
| **Currently smokes/sniffs/chews any tobacco products** | | | | | | |
| No | 676 | 244(35.8) | 54(8.0) | **0.001** | 162(24.0) | 0.884 |
| Yes | 213 | 85(39.9) | 33(15.5) | | 50(23.5) | |
| **Current smoke/sniff/chew tobacco products daily (N = 213)** | | | | | | |
| No | 9 | 1(11.1) | 2(22.2) | 0.197 | 3(33.3) | 0.476 |
| Yes | 204 | 84(41.2) | 31(15.2) | | 47(23.0) | |
| **Currently/in the past 12 months, consume any alcohol.** | | | | | | |
| No | 715 | 258(36.1) | 52(7.3) | **<0.001** | 152(21.3) | **<0.001** |
| Yes | 174 | 71(40.8) | 35(20.1) | | 60(34.5) | |
| **Frequency of alcohol consumption (N = 174)** | | | | | | |
| Two days or less/ week | 115 | 43(37.4) | 24(20.9) | 0.215 | 40(34.8) | 0.734 |
| At least days/week | 59 | 30(50.8) | 11(18.6) | | 19(32.2) | |
| **Eat fruit in a typical week.** | | | | | | |
| No | 426 | 163(38.3) | 42(9.9) | 0.732 | 98(23.0) | 0.572 |
| Yes | 463 | 166(35.9) | 45(9.7) | | 114(24.6) | |
| **Eat veg. in a typical week.** | | | | | | |
| No | 140 | 53(37.9) | 16(11.4) | 0.710 | 27(19.3) | 0.168 |
| Yes | 749 | 276(36.8) | 71(9.5) | | 185(24.7) | |
| **Frequency of adding salt to the food right before eating** | | | | | | |
| Never | 426 | 169(39.7) | 40(9.4) | 0.597 | 94(22.1) | 0.452 |
| Rarely | 219 | 78(35.6) | 21(9.6) | | 54(24.7) | |
| Sometimes | 244 | 82(33.6) | 26(10.7) | | 64(26.2) | |
| **Any physical activities** | | | | | | |
| No | 22 | 6(27.3) | 2(9.1) | 0.592 | 9(40.9) | 0.057 |
| Yes | 867 | 323(37.3) | 85(9.8) | | 203(23.4) | |
| **Family history of any CVDs from 1st-degree relative** | | | | | | |
| No | 848 | 309(36.4) | 81(9.6) | 0.087 | 195(23.0) | **0.007** |
| Yes | 41 | 20(48.8) | 6(14.6) | | 17(41.5) | |
| **BMI categories in kg/m$^2$** | | | | | | |
| Normal (18.5–24.9) | 616 | 231(37.5) | 55(8.9) | **<0.001** | 121(19.6) | **<0.001** |
| Underweight (≤18.4) | 132 | 40(30.3) | 6(4.5) | | 22(16.7) | |
| Overweight (25–29.9) | 108 | 46(42.6) | 17(15.7) | | 56(51.9) | |
| Obese (≥30) | 33 | 12(36.4) | 9(27.3) | | 13(39.4) | |
| **WHR categories (abnormal: >0.90(males) & >0.85(females)** | | | | | | |
| Normal | 387 | 140(36.2) | 33(8.5) | 0.406 | 75(19.4) | **0.006** |
| Abnormal | 502 | 189(37.6) | 54(10.8) | | 137(27.3) | |
| **The blood glucose level in mmol/L (RBG>11 or FBG>7 = Raised)** | | | | | | |
| Normal | 877 | 323(36.8) | 84(9.6) | 0.072 | 207(23.6) | 0.145 |
| Raised blood glucose | 12 | 6(50.0) | 3(25.0) | | 5(41.7) | |

*<120/80mmHg = Normal BP, 120/80-139.9/89.9mmHg = Elevated BP, ≥140/90mmHg = HTN

$\phi$<200mg/dL = Normal TC levels, ≥200mg/dL = HLP

not have any history of CVDs [(14.6% vs. 9.6%; p = 0.087) and (41.5% vs. 23.0%; p = 0.007), respectively]. In addition, participants who were overweight / obese had a significantly higher prevalence of both HTN and HLP as compared to their normal weight/non-obese

**Table 5. Ordinal and Bayesian logistic regression model for the effect of red meat consumption on HTN and HLP (N = 889).**

| Red meat consumption | HTN[φ] | | HLP[*] | |
|---|---|---|---|---|
| | PR (95% CI) | APR[a] (95% CI) | PR (95% CrI) [¥] | APR [b] (95% CrI) [¥] |
| **Amount consumed per week** | | | | |
| ≤250g/week | 1 | 1 | 1 | 1 |
| 251–750 (g/week) | 1.17 (0.85–1.61) | 1.04(0.74–1.46) | 1.20(0.79–1.75) | **1.56(1.10–2.09)** |
| >750g/week | 1.13(0.82–1.55) | 0.89(0.62–1.28) | 1.41(0.95–2.02) | **1.64(1.08–2.41)** |

[φ]Ordinal logistic regression was used to estimate the HTN prevalence ratio

[*]Bayesian logistic regression was used to estimate the HLP prevalence ratio

[ϛ]95%CI = 95% Confidence Intervals

[¥]95%CrI = 95% Credible Intervals

[a]Adjusted for: sex, age, education, marital status, occupation, monthly income, size of household, smoking status, alcohol consumption, blood ingestion, animal fats intake, salts intake, family history of CVDs, BMI, WHR, cholesterol level and glucose.

[b] Adjusted for: sex, age, household size, education level, marital status, occupation, monthly income, villages, alcohol consumption, animal fats intake, table salt intake, BMI and WHR

counterparts. Participants who had abnormal WHR had a significantly higher prevalence of HLP as compared to those who had normal WHR [27.3% vs. 19.4%, p = 0.006, respectively]. However, this difference was not statistically significant in HTN's prevalence [10.8% vs. 8.5%, p = 0.406, respectively] (**Table 4**).

## Association between HLP and HTN and red meat consumption

The findings from crude ordinal logistic regression analysis revealed that, the consumption of 251–750 grams of red meat per week (PR: 1.17, 95% CI 0.85–1.61) and >750 grams of red meat per week (PR: 1.13, 95% CI 0.82–1.55) was not statistically significantly associated with HTN. Similar results were observed from the Bayesian logistic regression model for HLP.

After controlling other factors, participants who reported to consume 251–750 (grams/week) (PR: 1.20, 95% CI 0.79–1.75) and >750 grams/week (PR: 1.41, 95%CrI 0.95–2.02) had significant prevalence of HLP as opposed to participants who reported to consume ≤250 grams/week of red meat. The association between red meat consumption and HTN remained insignificant, after adjusting for other factors (**Table 5**).

## Discussion

This study aimed at determining the prevalence of hypertension and hyperlipidaemia among the Maasai pastoralists of Ngorongoro Conservation Area in Arusha-Tanzania and to assess their association with the level of red meat consumption.

In this study, almost one-quarter of the participants were hyperlipidaemic and nearly 10percent were hypertensive. The prevalence of pre-hypertensive (elevated BP) was at least 3 times higher than that of hypertension. In both conditions, males were more vulnerable than were their female counterparts. Participants who reported to consume 251–750 grams and >750 grams of red meat per week had higher prevalence of HLP. On the other hand, there was no significant association between the consumption of red meat and the prevalence of HTN among the participants.

The prevalence of HTN observed in our study was not common among the study participants. This is in contrast to what was initially expected and what was consistent with the findings of previous study (see Ngoye et al. [27]. conducted in the same community. The similarity in the finding could be attributed to population stability in the study area. The prevalence of

HTN in the current study was lower than that reported in the earlier studies on the urban Maasai in Arusha town (27.7%), agro-pastoralists Maasai in Simanjiro District (21.4%) [7], and the national estimate (24.2%) [28]. The reason for the lower prevalence of HTN in our study could be attributed to the physically activeness of the study population. The majority of the participants were physically active thus reducing their vulnerability to HTN. The higher prevalence of HTN among the Maasai in Arusha urban and Simanjiro could be explained by the changes of lifestyle to cope with the new and urban cultural environment; this is unlike their counterparts residing in the NCA. The Maasai people in the former community are more exposed to different modern diets than are their counterparts in the NCA. The higher prevalence of HTN of 52percent has also been reported in China among adult minority from the pasture area [29]. The possible reasons for the difference between our study and the Chinese study in the prevalence of HTN could be explained by the differences in dietary related eating habits between the two populations. The Chinese communities prefer to consume excessive alcohol and salted foods that lead to increased blood pressure. This is in contrast to the NCA Maasai where majority (80%) are not consuming alcohol and prepare their food such as meat in a traditional way without adding salt. In the present study, the prevalence of HTN was higher among males than was among their female counterparts. But the levels were lower than those reported in the previous studies among rural residents in Shari, Kilimanjaro region (32.2% vs. 31.5%, for male and female participants, respectively) [30]. As noted by the previous authors, the possible explanation for the high prevalence of HTN in the previous studies could be attributed to the adaptation of sedentary lifestyle [30]. However, high prevalence of elevated BP in the present study calls for a need of revising strategies of shifting the BP among the population to the left direction because if the current situation is left unattended, it might increase the risk of developing HTN among this population.

The prevalence of HLP in our study was slightly higher than the national estimates (22.1%) [28], but it was lower than was the levels reported in the previous study in China (51.5%) [29]. The elevated prevalence of HLP in the current study could be attributed to high consumption of red meat above the recommended threshold (500g/week) for the Tanzania population [31]. It was clear from this study that, the majority (99.4%) of the participants in the study community reported to be big consumers of red meat; thus, majority (69.5%) reported a high intake of animal fats. In addition, numerous modified risk factors such as tobacco use, alcohol consumption, table salt intake, abnormal WHR, and overweight/obesity were predominant among the people in this community. These factors have been associated with an increase of the risk of HLP [32]. The difference between our study and the Chinese study (49.2%)[29] in the prevalence of HLP could be attributed to excessive consumptions of alcohol, excessive intake of animal fats and salted foods among the Chinese. This was not the case in the present study. The low serum lipids levels have also been reported among pastoral communities in some African and European countries such as Fulani in Nigeria and herders in Siberia respectively. None of these reported the prevalence of HLP [33,34]. As for HTN, the prevalence of HLP in the study population was higher among males than was among their female counterparts (29.0% vs. 20.1%, respectively). This observation was consistent with the observation in a previous study conducted among the Maasai in Monduli District, Tanzania [35]. This finding may be explained by similarities in population characteristics between the study areas. As revealed by the present study, unlike women, the Maasai men prefer to consume red meat and blood. This finding might be the case in the previous study since both studies were conducted to a similar population with shared cultural values. Previously, studies demonstrated a gender difference among the Maasai, where as opposed to women; men stay closer to their livestock (caretakers) that give them the advantage of consuming animal products. The responsibilities of women include building houses, fetching water and performing other domestic chores [36].

These compel them to stay at home and thus rarely change their diet. Previous authors also revealed that the Maasai women of reproductive age are recommended to consume low energy food, fat, and protein to avoid a large birth weight and birth complication [37]. All these might explain the observed differences in the prevalence of HLP between the current study and other studies in the study area.

Red meat consumption was positively associated with the high prevalence of HLP among the NCA Maasai. In Comparison to participants consuming ≤250 grams/week, the prevalence of HLP was higher among the participants consuming between 251–750 grams/week and >750 grams/week respectively. A similar finding is reported elsewhere [38]. This finding could be a reflection of higher rate of red meat consumption in the study community where most (99.4%) of the study participants particularly men reported to be consuming red meat excessively. The observed association between high-level of red meat consumption and increased prevalence of HLP in the study population calls for a need of designing strategies of reverse the trend of HLP prevalence and thereby minimizing the risk for CVDs problem. the current government initiatives including the provision of free food to the Maasai community living in the NCAA need to be promoted because it offers the opportunity of food diversification [18]. This practice will prevent the community from being entirely dependent on animal products for food [39]. It is worth noting that remoteness nature of the study population, dependence on red meat, and migratory patterns reduce the community's access to health systems, making diagnosis and management of these conditions become a challenge.

There was no statistically significant association between red meat consumption and HTN in this study. This finding is in contrast with the finding in the previous studies in France, Iran, and Sweden [40–42]. The possible reasons of these differences could be attributed to the predominance of high levels of physical activities (97.4%) among the NCA Maasai and the use of additive plants (traditional herbs), which could probably be offering protection against HTN [39]. However, the protective effect of additive plants against HTN remains a speculation, which calls for further studies to establish its validity.

## Limitation of the study

Despite important information generated by this study, our results should be interpreted with caution due to many limitations. The interview process used a self-reporting method. This approach is prone to social desirability bias; therefore, it may overestimate the actual amount of red meat consumed. The use of Tanzania Food Composition Table to estimate the amount consumed is prone to measurement bias, which may over or underestimate the actual amount eaten vis-a-vis the recorded weight. The use of total cholesterol as an overall measure of HLP was challenging in showing specific levels of HDL, LDL, and TG in the blood. This was due to the limited funding available during the conduct of this study.

## Conclusion

The prevalence of HLP was high among the participants in the study area. Although the prevalence of HTN was not common, the high prevalence of elevated BP calls for an agent need for early intervention such as a change of lifestyle patterns as well as awareness creation on the risk factors of HTN. The consumption of red meat was associated with an increase in the probability of getting HLP. Therefore, efforts of shifting the distribution of HLP to the left direction in this population, including reduction of red meat consumption are warranted.

## Supporting information

**S1 File.**
(DOCX)

**S2 File.**
(DOCX)

## Acknowledgments

The authors are highly grateful to the participants of this study. Without their hospitality and cooperation, this study would not have been possible. Special thanks should go to the Regional Administrative Secretary, District Administrative Secretary, District Medical Officer, Endulen Hospital Doctor In-charge, Ngorongoro Conservation Area Authority, Wards Executive Officers, and Village leaders for granting me permission to conduct this study in their areas. We are also grateful to the Staff of Epidemiology and Applied Biostatistics Department of KCMUCo (Dr Rune Philemon, Dr Damian Jeremiah Damian, Mr Innocent Mboya, and Ms Melina Mgongo) for their comments and technical support. Lastly, we appreciate the efforts and time devoted by the research assistants during the data collection period.

## Author Contributions

**Conceptualization:** Ester J. Diarz, Beatrice J. Leyaro, Sokoine L. Kivuyo, Bernard J. Ngowi, Sia E. Msuya, Sayoki G. Mfinanga, Michael J. Mahande.

**Data curation:** Ester J. Diarz, Beatrice J. Leyaro, Sia E. Msuya, Michael J. Mahande.

**Formal analysis:** Ester J. Diarz.

**Funding acquisition:** Bassirou Bonfoh.

**Investigation:** Ester J. Diarz.

**Methodology:** Ester J. Diarz, Bernard J. Ngowi, Sia E. Msuya, Bassirou Bonfoh, Michael J. Mahande.

**Project administration:** Ester J. Diarz.

**Software:** Ester J. Diarz.

**Supervision:** Bernard J. Ngowi, Sia E. Msuya, Sayoki G. Mfinanga, Bassirou Bonfoh, Michael J. Mahande.

**Validation:** Michael J. Mahande.

**Writing – original draft:** Ester J. Diarz.

**Writing – review & editing:** Beatrice J. Leyaro, Sokoine L. Kivuyo, Bernard J. Ngowi, Sia E. Msuya, Sayoki G. Mfinanga, Bassirou Bonfoh, Michael J. Mahande.

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
