## [Decision Letter · Decision Letter 0]

3 Feb 2020

PONE-D-19-31017

RED MEAT CONSUMPTION AND ITS ASSOCIATION WITH HYPERTENSION AND HYPERLIPIDEMIA AMONG ADULT MAASAI PASTORALISTS IN NGORONGORO CONSERVATION AREA, TANZANIA

PLOS ONE

Dear Ms DIARZ,

Thank you for submitting your manuscript to PLOS ONE. After careful consideration, we feel that it has merit but does not fully meet PLOS ONE’s publication criteria as it currently stands. Therefore, we invite you to submit a revised version of the manuscript that addresses the points raised during the review process.

We would appreciate receiving your revised manuscript by Mar 19 2020 11:59PM. To enhance the reproducibility of your results, we recommend that if applicable you deposit your laboratory protocols in protocols.io, where a protocol can be assigned its own identifier (DOI) such that it can be cited independently in the future. For instructions see: http://journals.plos.org/plosone/s/submission-guidelines#loc-laboratory-protocols

We look forward to receiving your revised manuscript.

Kind regards,

Venkata Naga Srikanth Garikipati, PhD

Academic Editor

PLOS ONE

Additional Editor Comments (if provided):

This manuscript needs to be revised according to the reviewers comments

Journal Requirements:

2. Please include additional information regarding the survey or questionnaire used in the study and ensure that you have provided sufficient details that others could replicate the analyses. For instance, if you developed a questionnaire as part of this study and it is not under a copyright more restrictive than CC-BY, please include a copy, in both the original language and English, as Supporting Information. Moreover, please include more details on how the questionnaire was pre-tested, and whether it was validated.

4.Please include captions for your Supporting Information files at the end of your manuscript, and update any in-text citations to match accordingly. Please see our Supporting Information guidelines for more information: http://journals.plos.org/plosone/s/supporting-information.

Reviewers' comments:

Reviewer's Responses to Questions

**Comments to the Author**

1. Is the manuscript technically sound, and do the data support the conclusions?

Reviewer #1: Yes

Reviewer #2: Partly

2. Has the statistical analysis been performed appropriately and rigorously? 

Reviewer #1: Yes

Reviewer #2: Yes

3. Have the authors made all data underlying the findings in their manuscript fully available?

Reviewer #1: Yes

Reviewer #2: Yes

4. Is the manuscript presented in an intelligible fashion and written in standard English?

Reviewer #1: Yes

Reviewer #2: Yes

5. Review Comments to the Author

Reviewer #1: The manuscript title “Red meat consumption and its association with hypertension and hyperlipidemia among adult maasai pastoralists in ngorongoro conservation area, Tanzania” is well study and shown association with hypertension and hyperlipidaemia among Maasai pastoralists of Ngorongoro Conservation Area in Arusha-Tanzania with the level of red meat consumption. It’s well report and well report.

Reviewer #2: The authors in this manuscript provide a population based study of the Tanzanian Maasai community's red meat consumption. -Owing to the limitation in the socio-economic status of this community, their lifestyle governs a lot on their food habits. -Having excess red meat consumption leads to hyperlipidemia.

This work can help in providing people of that region an insight to develop a method of proper balance diet to combat cardiovascular diseases.

The authors could have tried to overcome the limitation of not defining the lipid profile in details, by use of a simple diagnostic kit used these days for biometric screening. However, I still would like to accept this manuscript for publication as this can help the developing countries raise their dietary standards along with WHO guidance.

6. PLOS authors have the option to publish the peer review history of their article (what does this mean?). If published, this will include your full peer review and any attached files.

Reviewer #1: No

Reviewer #2: No

---

## [Author Response · Author response to Decision Letter 0]

18 Mar 2020

We sincerely thanks to all reviewers comments and their time to review our manuscript. Below are the responses regarding comments raised by both editor and reviewers.

Editor Comments:

 RESPONSE: The manuscript followed the PLOS ONE’s style requirements.

2. Please include additional information regarding the survey or questionnaire used in the study and ensure that you have provided sufficient details that others could replicate the analyses. For instance, if you developed a questionnaire as part of this study and it is not under a copyright more restrictive than CC-BY, please include a copy, in both the original language and English, as Supporting Information. Moreover, please include more details on how the questionnaire was pre-tested, and whether it was validated.

RESPONSE: Data collection tool was adopted from WHO Non-Communicable Diseases Stepwise approach to surveillance (STEPS) https://www.who.int/ncds/surveillance/steps/en/ This was then modified with additional questions to account for cultural diet of Maasai in Tanzania (attached in supporting information in both Swahili and English language). The questionnaire included; social demographic information (age gender, income etc), data on life style (physical activity, tobacco use, alcohol consumption, dietary intake) and relevant family history information was collected from each participant through self-reporting.Training of the data collection activity was conducted for five days. Pre-test of the tools and procedures were done in 4th day of training to the community around Kilimanjaro Christian Medical University College area. After the pre-test, feedback was shared, challenges observed from the pre-test were explained and finalization of the tools were done.

 RESPONSE: No changes of the data availability statement.

4.Please include captions for your Supporting Information files at the end of your manuscript, and update any in-text citations to match accordingly. Please see our Supporting Information guidelines for more information: http://journals.plos.org/plosone/s/supporting-information.

RESPONSE: Captions of the supporting information are included in the manuscript.

Reviewers' comments:

RESPONSE: We thank reviewer 1 and 2 for positive comments regarding different sections of our manuscript, specifically with regard on question 1, 2, 3, 4 and 6 as there is no action needed. Below is our response to reviewer # 2 comments on question 5.

5. Review Comments to the Author

Reviewer #2: The authors in this manuscript provide a population-based study of the Tanzanian Maasai community's red meat consumption. -Owing to the limitation in the socio-economic status of this community, their lifestyle governs a lot on their food habits. -Having excess red meat consumption leads to hyperlipidemia.

This work can help in providing people of that region an insight to develop a method of proper balance diet to combat cardiovascular diseases.

The authors could have tried to overcome the limitation of not defining the lipid profile in details, by use of a simple diagnostic kit used these days for biometric screening. However, I still would like to accept this manuscript for publication as this can help the developing countries raise their dietary standards along with WHO guidance.

RESPONSE: We completely agree with the reviewer comment on these. However, due to the limited funding available during the conduct of this study we couldn’t explore much on lipids. We believe this study will provide data for further robust studies to explore in details the association of red meat and lipids as well as hypertension in this Masai community.

---

## [Decision Letter · Decision Letter 1]

13 May 2020

RED MEAT CONSUMPTION AND ITS ASSOCIATION WITH HYPERTENSION AND HYPERLIPIDEMIA AMONG ADULT MAASAI PASTORALISTS IN NGORONGORO CONSERVATION AREA, TANZANIA

PONE-D-19-31017R1

Dear Dr. DIARZ,

We are pleased to inform you that your manuscript has been judged scientifically suitable for publication and will be formally accepted for publication once it complies with all outstanding technical requirements.

With kind regards,

Venkata Naga Srikanth Garikipati, PhD

Academic Editor

PLOS ONE

Additional Editor Comments (optional):

Reviewers' comments:

Reviewer's Responses to Questions

**Comments to the Author**

1. If the authors have adequately addressed your comments raised in a previous round of review and you feel that this manuscript is now acceptable for publication, you may indicate that here to bypass the “Comments to the Author” section, enter your conflict of interest statement in the “Confidential to Editor” section, and submit your "Accept" recommendation.

Reviewer #1: All comments have been addressed

Reviewer #2: All comments have been addressed

2. Is the manuscript technically sound, and do the data support the conclusions?

Reviewer #1: Yes

Reviewer #2: Yes

3. Has the statistical analysis been performed appropriately and rigorously? 

Reviewer #1: Yes

Reviewer #2: Yes

4. Have the authors made all data underlying the findings in their manuscript fully available?

Reviewer #1: Yes

Reviewer #2: Yes

5. Is the manuscript presented in an intelligible fashion and written in standard English?

Reviewer #1: Yes

Reviewer #2: Yes

6. Review Comments to the Author

Reviewer #1: To

Author

For the Manuscript title “Red meat consumption and its association with hypertension and hyperlipidemia among adult maasai pastoralists in ngorongoro conservation area, tanzania. Addressed all the reviewer comments and improved . It’s can be accept

Thanks

Reviewer #2: The authors have responded as per the comments in their best capabilities. Although, I understand that few experiments were not conducted owing to their funding issue.

7. PLOS authors have the option to publish the peer review history of their article (what does this mean?). If published, this will include your full peer review and any attached files.

Reviewer #1: No

Reviewer #2: No

---

## [Editor Report · Acceptance letter]

22 May 2020

PONE-D-19-31017R1 

Red meat consumption and its association with hypertension and hyperlipidaemia among adult Maasai pastoralists of Ngorongoro Conservation Area, Tanzania 

Dear Dr. Diarz:

I am pleased to inform you that your manuscript has been deemed suitable for publication in PLOS ONE. Congratulations! Your manuscript is now with our production department. 

With kind regards,

on behalf of

Dr. Venkata Naga Srikanth Garikipati 

Academic Editor

PLOS ONE